# Central Dicer-miR-103/107 controls developmental switch of POMC progenitors into NPY neurons and impacts glucose homeostasis

Sophie Croizier[1,2†], Soyoung Park[1,2], Julien Maillard[1,2,3], Sebastien G Bouret[1,2,3]*

[1]The Saban Research Institute, Developmental Neuroscience Program, Diabetes and Obesity Program, Center for Endocrinology, Diabetes and Metabolism, Children's Hospital Los Angeles, Los Angeles, United States; [2]Pediatrics, University of Southern California, Los Angeles, California; [3]Jean-Pierre Aubert Research Center, Inserm U1172, Lille 2 University of Health and Law, Lille, France

**Abstract** Proopiomelanocortin (POMC) neurons are major negative regulators of energy balance. A distinct developmental property of POMC neurons is that they can adopt an orexigenic neuropeptide Y (NPY) phenotype. However, the mechanisms underlying the differentiation of *Pomc* progenitors remain unknown. Here, we show that the loss of the microRNA (miRNA)-processing enzyme *Dicer* in POMC neurons causes metabolic defects, an age-dependent decline in the number of *Pomc*mRNA-expressing cells, and an increased proportion of *Pomc* progenitors acquiring a NPY phenotype. miRNome microarray screening further identified miR-103/107 as candidates that may be involved in the maturation of *Pomc* progenitors. In vitro inhibition of miR-103/107 causes a reduction in the number of *Pomc*-expressing cells and increases the proportion of *Pomc* progenitors differentiating into NPY neurons. Moreover, in utero silencing of miR-103/107 causes perturbations in glucose homeostasis. Together, these data suggest a role for prenatal miR-103/107 in the maturation of *Pomc* progenitors and glucose homeostasis.
DOI: https://doi.org/10.7554/eLife.40429.001

*For correspondence:
sbouret@chla.usc.edu

Present address: †Center for Integrative Genomics, University of Lausanne, Lausanne, Switzerland

Competing interests: The authors declare that no competing interests exist.

## Introduction

The growing prevalence of obesity is a major health concern, particularly among children. Recent evidence has shown that obesity and associated diseases, such as type II diabetes, might be a consequence of alterations in the developmental processes of a variety of systems involved in energy balance regulation, including the hypothalamic melanocortin system (for review see (*Bouret et al., 2015*; *Ralevski and Horvath, 2015*; *Steculorum et al., 2013*). An important component of this system includes pro-opiomelanocortin (POMC) neurons and neurons co-expressing agouti-related protein (AgRP) and neuropeptide Y (NPY) in the arcuate nucleus of the hypothalamus (ARH) (*Gautron et al., 2015*; *Koch and Horvath, 2014*). POMC neurons reduce food intake and increase energy expenditure by releasing α-melanocyte-stimulating hormone (αMSH), a product of POMC processing that activates melanocortin-4 receptors (MC4R) (*Cone, 2006*; *Koch and Horvath, 2014*). AgRP acts as an endogenous inverse agonist of MC4R.

The developmental processes that produce POMC neuronal pathways can be divided into two broad categories: the determination of cell numbers, which involves neurogenesis, neuron migration, and the determination of neuronal phenotype, and the formation of functional circuits, which includes axon growth and synaptogenesis (see (*Markakis, 2002*; *Coupe and Bouret, 2013*) for review). Birthdating studies in rodents indicate that POMC neurons in the ARH are generated

primarily on embryonic day (E) 11-E12 and acquire their terminal peptidergic phenotype during mid-gestation (*Padilla et al., 2010*). Remarkably, cell lineage tracing strategies showed that a subpopulation of embryonic *Pomc*-expressing precursors can differentiate into NPY neurons (*Padilla et al., 2010*). A second important developmental period occurs during the first few weeks of postnatal life when POMC neurons send axonal projections to their target sites. The development of axonal projections from the ARH (including αMSH-containing projections) to each of their target nuclei is not initiated until the first week of postnatal life, and these projections are not fully mature until weaning in mice (*Bouret et al., 2004a*).

The process of developing highly specialized cellular structures, such as POMC neurons, requires the tight temporal and regional regulation of expression for specific sets of genes. However, the molecular mechanisms underlying POMC neuronal development still remain elusive. MicroRNAs (miRNAs) have recently emerged as critical regulators of brain development. These non-coding small endogenous RNA molecules have important functions in gene regulation. They bind to complementary sequences on protein-coding mRNAs and consequently direct their degradation and/or repress their translation (*Bartel, 2009*). miRNAs exhibit cell- and tissue-specific and developmental stage-specific expression (*Lagos-Quintana et al., 2002*). They play important roles in a variety of biological programs, including neuronal differentiation, neuronal survival, axon growth, synaptogenesis and other developmental processes (*Stefani and Slack, 2008*). *Dicer* is one of the essential enzyme for miRNA maturation (*Fineberg et al., 2009*). *Dicer* ablation in various specific populations of neurons has been shown to impair cell fate specification, cause neuronal cell death, and disrupt axon growth (*Fineberg et al., 2009*; *Vo et al., 2010*).

Although recent advances have indicated that miRNAs are important regulators of neural development, the role of these non-coding small endogenous RNA molecules in the development of neural systems involved in energy balance regulation remains unclear. In the present study, we investigated the role of miRNAs in the phenotypic differentiation of *Pomc* progenitors. Our findings revealed that miRNAs are essential for survival and timely maturation of POMC neurons and that loss of *Dicer* favors the differentiation of *Pomc*-expressing progenitor cells into NPY neurons and causes metabolic perturbations. We also showed that miR-103/107 play a particularly important role in these processes.

## Results

### Loss of *Dicer* in POMC neurons causes metabolic dysregulation

To examine whether miRNAs play a role in hypothalamic development, we first measured *Dicer* mRNA expression, an essential enzyme for miRNA maturation (*Fineberg et al., 2009*), in the embryonic, postnatal and adult hypothalamus. The highest levels of *Dicer* mRNA were found in hypothalamus of mice at embryonic day (E) 14 and 16, supporting a role for miRNAs in embryonic hypothalamic development (*Figure 1a*). *Dicer* mRNA levels decreased at postnatal day (P) 10 and the lowest levels of *Dicer* mRNA were found in the hypothalamus of adult mice (*Figure 1a*). We next assessed *Dicer* mRNA expression specifically in POMC neurons and found that *Dicer* mRNA was expressed in isolated POMC neurons as early as at E13-E15, that is when *Pomc* progenitor cells differentiate to either POMC or NPY neurons (*Figure 1b*). Notably, *Dicer* mRNA was also highly expressed in NPY neurons at E15 (*Figure 1b*).

To determine whether Dicer is required for the normal development of POMC neurons in vivo, we crossed mice carrying a *Dicer*[loxP] allele (*Harfe et al., 2005*) with mice expressing *Cre* recombinase in a *Pomc*-specific manner (*Pomc*-Cre) (*Balthasar et al., 2004*). The resulting *Pomc*-Cre; *Dicer*[loxP/loxP] male mice displayed significantly higher body weights compared with control *Dicer*[loxP/loxP] mice starting at 6 weeks of age (*Figure 1c and d*). In addition, *Pomc*-Cre; *Dicer*[loxP/loxP] male mice displayed reduced oxygen consumption ($VO_2$), carbon dioxide production ($VCO_2$), respiratory quotient, energy expenditure, and locomotor activity (*Figure 1e–g*). In addition, mutant male mice had higher food consumption during dark phase (*Figure 1h*). Epididymal fat pad weights were also significantly higher in adult *Pomc*-Cre; *Dicer*[loxP/loxP] male mice compared with *Dicer*[loxP/loxP] mice (*Figure 1i*). Serum leptin levels were also significantly elevated in mutant male mice compared with control mice (*Figure 1j*). In addition, we performed leptin sensitivity tests and observed that the weight loss effect of leptin was attenuated in *Pomc*-Cre; *Dicer*[loxP/loxP] male mice compared with

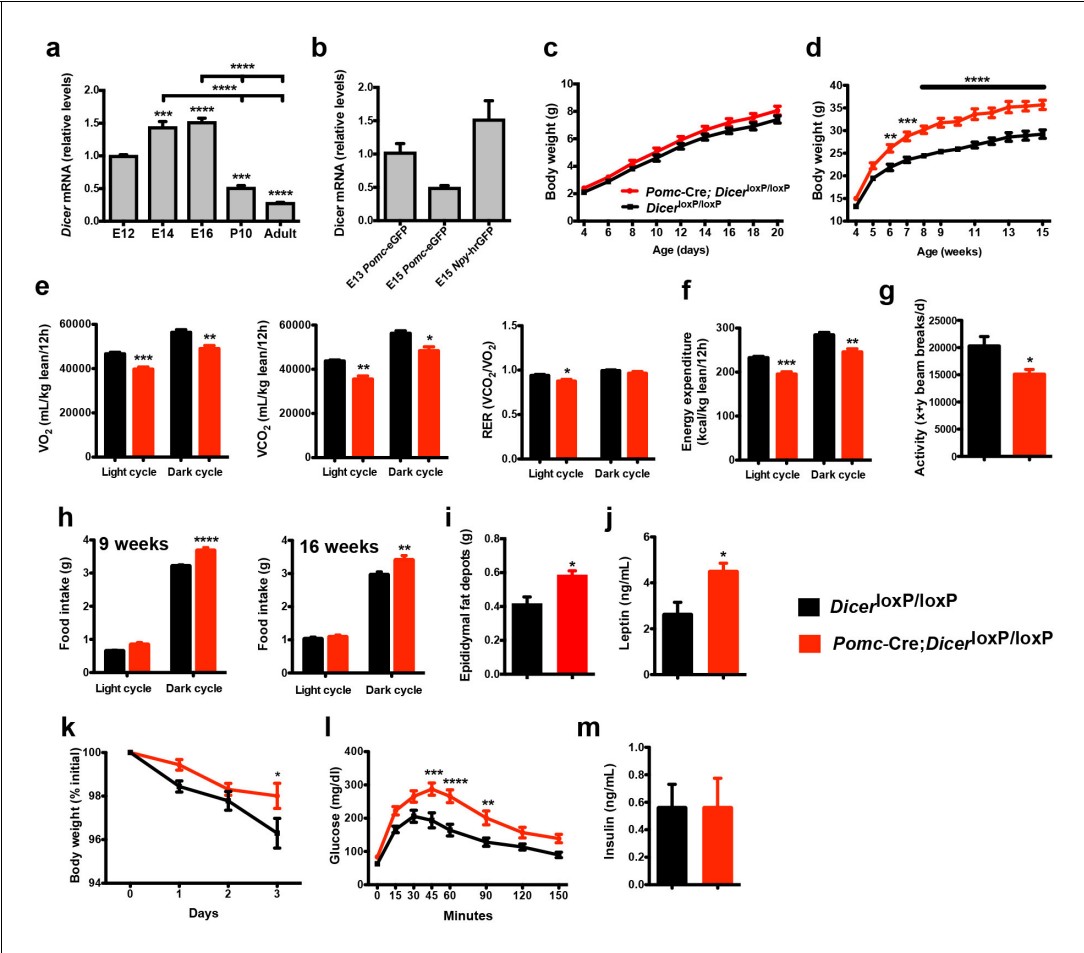

**Figure 1.** Loss of *Dicer* in *Pomc* expressing neurons causes metabolic dysregulation. (a) Relative expression of *Dicer* mRNA in the hypothalamus of E12, E14, E16 embryos and in the mediobasal hypothalamus of P10, and adult mice (n = 3 – 5 per group). (b) Relative expression of *Dicer* mRNA in sorted *Pomc*-eGFP cells at E13 and E15 and *Npy*-hrGFP cells at E15 (n = 2 – 4 per group). (c) Pre- and (d) post-weaning growth curves (body weights) of *Dicer*$^{loxP/loxP}$ and *Pomc*-Cre; *Dicer*$^{loxP/loxP}$ male mice (n ≥ 8 per group). (e) $VO_2$, $VCO_2$ and respiratory exchange ratio (RER) of 15-week-old *Dicer*$^{loxP/loxP}$ and *Pomc*-Cre; *Dicer*$^{loxP/loxP}$ male mice (n = 4 – 6 per group). (f) Energy expenditure of 15-week-old *Dicer*$^{loxP/loxP}$ and *Pomc*-Cre; *Dicer*$^{loxP/loxP}$ male mice (n = 4 – 6 per group). (g) Locomotor activity of 15-week-old *Dicer*$^{loxP/loxP}$ and *Pomc*-Cre; *Dicer*$^{loxP/loxP}$ male mice (n = 4 – 6 per group). (h) Cumulative food intake of 9- and 16-week-old *Dicer*$^{loxP/loxP}$ and *Pomc*-Cre; *Dicer*$^{loxP/loxP}$ male mice (n = 5 – 6 per group). (i) Epididymal fat mass of 20- to 23-week-old male mice (n = 4 – 5 per group). (j) Plasma leptin levels in 20- to 23-week-old *Dicer*$^{loxP/loxP}$ and *Pomc*-Cre; *Dicer*$^{loxP/loxP}$ male mice (n = 4 – 5 per group). (k) Leptin-induced weight loss in 14-week-old *Dicer*$^{loxP/loxP}$ and *Pomc*-Cre; *Dicer*$^{loxP/loxP}$ male (n = 5 per group). (l) Glucose tolerance test of 10- to 11-week-old male mice (n ≥ 7 per group). (m) Plasma insulin levels in 20- to 23-week-old *Dicer*$^{loxP/loxP}$ and *Pomc*-Cre; *Dicer*$^{loxP/loxP}$ male mice (n = 6 per group). Values are shown as mean ±SEM. *p≤0.05 *versus Dicer*$^{loxP/loxP}$ (e, g, i, j); **p≤0.01 *versus Dicer*$^{loxP/loxP}$ (d, e, f, h, l); ***p≤0.001 *versus* E12 WT (a), *versus Dicer*$^{loxP/loxP}$ (d, e, f, l); ****p≤0.0001 *versus* E12 WT (a), *versus* E14 WT (a), *versus* E16 WT (a), *versus Dicer*$^{loxP/loxP}$ (d, l). Statistical significance was determined using 2-tailed Student's *t* test (e, f, g, i, j, m), 1-way ANOVA followed by Turkey's *post hoc* test (a, b) and 2-way ANOVA followed by Bonferroni's *post hoc* test (c, d, h, k, l).

DOI: https://doi.org/10.7554/eLife.40429.002

The following figure supplement is available for figure 1:

**Figure supplement 1.** Altered metabolism in female mice lacking *Dicer* in *Pomc*-expressing neurons.

DOI: https://doi.org/10.7554/eLife.40429.003

*Dicer*$^{loxP/loxP}$ mice (*Figure 1k*). In addition, when exposed to a glucose challenge, mutant male mice displayed impaired glucose tolerance compared with control mice (*Figure 1l*). However, basal insulin levels remained unchanged (*Figure 1m*). Most of the metabolic abnormalities were also found in female mice, including increased body weight, food intake, fat mass and reduced $VO_2$, $VCO_2$, and energy expenditure (*Figure 1—figure supplement 1*). Adult female mutant mice also displayed impaired glucose tolerance compared with control mice (*Figure 1—figure supplement 1f*).

Together, these data indicate that the loss of *Dicer* in POMC neurons has functional consequences on energy balance and glucose regulation.

## Loss of *Dicer* in POMC neurons is associated with a marked reduction in the number of *Pomc* mRNA-expressing cells

POMC neurons in the ARH are generated primarily on embryonic day (E) 11-E12 and acquire their terminal peptidergic phenotype during mid-late gestation (*Padilla et al., 2010*). Because miRNAs have recently emerged as critical regulators of brain development and *Dicer* mRNA is expressed in POMC neurons during important periods of neurogenesis and cell fate, we examined whether lack of miRNA maturation in POMC neurons causes neurodevelopmental alterations. We first performed in situ hybridization (ISH) experiments and counted the number of neurons expressing *Pomc* mRNA in the ARH of *Pomc*-Cre; *Dicer*loxP/loxP and *Dicer*loxP/loxP mice. The number of *Pomc* mRNA-expressing cells in the ARH of *Pomc*-Cre; *Dicer*loxP/loxP embryos at E13 and E15 was 1.4- and 1.2-fold lower, respectively, than that observed in control embryos (*Figure 2a*). *Pomc*-Cre; *Dicer*loxP/loxP mice P10 and P14 displayed a 1.3- and 1.5-fold reduction, respectively, in the number of *Pomc* mRNA-expressing cells (*Figure 2a*; *Figure 2—figure supplement 1a*). At weaning (P21) and in 15-week-old animals there was an 8.0- and 8.7-fold reduction, respectively, in the number of *Pomc* mRNA-expressing cells between mutant and control mice (*Figure 2a*). This marked reduction in the number of *Pomc* mRNA-expressing cells was accompanied by a decrease in *Pomc* mRNA content in the hypothalamus of P21 and 15-week-old mice (*Figure 2b*). In addition, a 3.7-fold reduction in the number of β-endorphin-immunoreactive cells (a peptide produced from POMC) was found in the ARH of *Pomc*-Cre; *Dicer*loxP/loxP mice (*Figure 2—figure supplement 1b*). In addition, the density of α-MSH- and β-endorphin-immunoreactive fibers was 7.3- and 11.3-fold decreased, respectively, in the magnocellular part of the PVH and 5.7- and 10.3-fold decreased, respectively, in the parvicellular part of the PVH of mutant mice (*Figure 2d*).

This dramatic reduction in the number of POMC-containing cells and fibers in the absence of *Dicer* could be caused by an increase in cell death as previously reported in various neuronal systems, including the hypothalamus (*Sundermeier et al., 2014*; *Zehir et al., 2010*; *Damiani et al., 2008*; *Schneeberger et al., 2012*). It is also possible that some cells survive but that *Pomc* gene expression is blunted in *Pomc*-Cre; *Dicer*loxP/loxP mice. To test this hypothesis, we used a cell lineage approach and crossed our *Pomc*-Cre mouse line with a *ROSA-tdTomato* reporter line to genetically and permanently label *Pomc* progenitor cells, independently of *Pomc* mRNA content. Despite a marked reduction in the number of *Pomc* mRNA-expressing cells at E15 (*Figure 2a*), the number of tdTomato⁺ cells was identical between *Pomc*-Cre; *Dicer*loxP/loxP and *Dicer*loxP/loxP mice at E15 (*Figure 2c*). At P14 (*Figure 2—figure supplement 1d*) and weaning (P21, *Figure 2c*), we observed a 1.9- and 2.9-fold reduction, respectively, in the number of tdTomato⁺ cells in mutant mice. The number of tdTomato⁺ cells found in the ARH was 4-fold reduced at 15 weeks of age (*Figure 2c*) but this reduction was not as marked as the decrease in the number of *Pomc* mRNA-expressing cells that we observed using in situ hybridization (8.7-fold, *Figure 2a*). These observations indicate that this reduction in POMC cell numbers is caused by a decrease in *Pomc* detectability *versus* cell loss. In addition, a significant reduction in POMC-immunoreactive cell numbers was observed in the pituitary of mutant mice but no change in tdTomato +cell numbers was found in the NTS, another brain site that contains POMC neurons (*Figure 2—figure supplement 1e and f*). These observations support the idea that loss of *Dicer* in hypothalamic *Pomc* progenitor cells causes postnatal cell death but also blunts *Pomc* gene expression and might favor acquisition of a non-POMC phenotype.

## Loss of dicer in POMC neurons favors the differentiation of *Pomc*-expressing progenitors into NPY neurons

Our data indicate a portion of genetically labeled *Pomc*-tdTomato⁺ cells do not express *Pomc* mRNA in *Pomc*-Cre; *Dicer*loxP/loxP mice. Because *Pomc* progenitors are known to adopt a POMC but also a non-POMC phenotype such as NPY (*Padilla et al., 2010*), we next quantified the number of NPY neurons in the ARH of control and mutant mice. First we performed in situ hybridizations and counted the number of *Npy* mRNA-expressing neurons and found that it was identical between *Pomc*-Cre; *Dicer*loxP/loxP and *Dicer*loxP/loxP mice at E15, P14, and P21 (*Figure 3a*; *Figure 2—figure supplement 1c*). Consistent with these findings, RT-qPCR experiments indicated that *Npy* mRNA

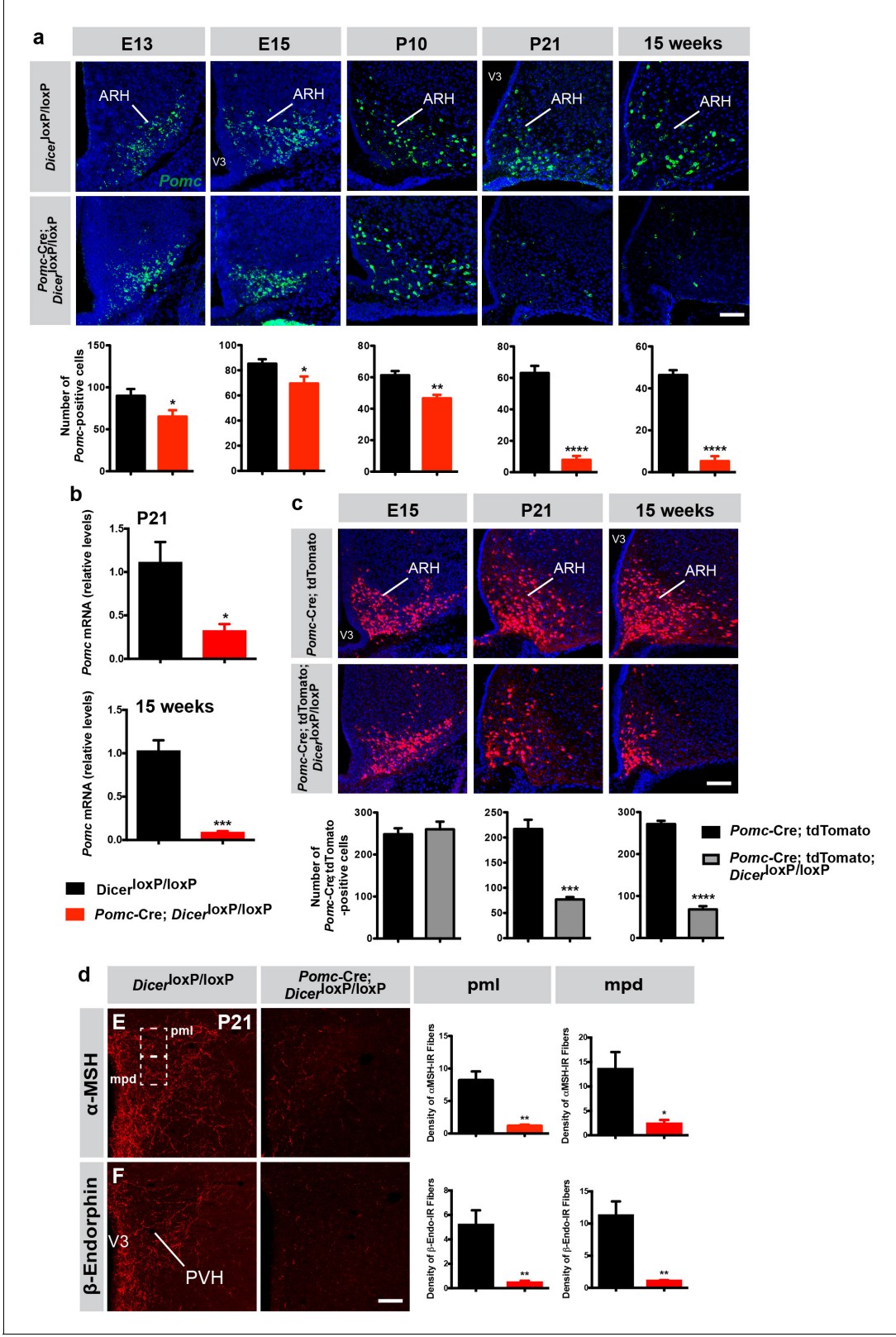

**Figure 2.** Reduced number of *Pomc* mRNA-expressing cells in mice lacking *Dicer* in POMC neurons. (a) Representative images and quantification of *Pomc* mRNA-expressing cells in the arcuate nucleus (ARH) of E13 (n = 6 per group), E15 (n = 6 – 8 per group), P10 (n = 4 per group), P21 (n = 4 per group) and 15-week-old (n = 3 – 4 per group) *Dicer*loxP/loxP and *Pomc*-Cre; *Dicer*loxP/loxP male mice. (b) Relative expression of *Pomc* mRNA in the mediobasal hypothalamus of P21 and 15-week-old *Dicer*loxP/loxP and *Pomc*-Cre; *Dicer*loxP/loxP male mice (n = 3 – 4 per group). (c) Representative images

*Figure 2 continued on next page*

*Figure 2 continued*

and quantification of *Pomc-Cre* cells genetically labeled by tdTomato in the ARH of E15 (n = 5 – 7 per group), P21 (n = 4 per group) and 15-week-old (n = 3 – 4 per group) *Pomc*-Cre and *Pomc*-Cre; *Dicer*$^{loxP/loxP}$ male mice. (d) Representative images and quantification of the density of α-MSH- and β-endorphin-immunopositive fibers (n = 3 – 4 per group) in the posterior magnocellular (pml) and medial parvicellular (mpd) parts of the paraventricular nucleus (PVH) of P21 *Dicer*$^{loxP/loxP}$ and *Pomc*-Cre; *Dicer*$^{loxP/loxP}$ male mice. Scale bars, 100 μm (a, b, c). Values are shown as mean ± SEM. *p≤0.05 *versus Dicer*$^{loxP/loxP}$ (a, b, d); **p≤0.01 *versus Dicer*$^{loxP/loxP}$ (a, d); ***p≤0.001 *versus Dicer*$^{loxP/loxP}$ (b), *versus Pomc*-Cre; tdTomato (c); ****p≤0.0001 *versus Dicer*$^{loxP/loxP}$ (a), *versus Pomc*-Cre; tdTomato (c). Statistical significance was determined using 2-tailed Student's *t* test. V3, third ventricle.

DOI: https://doi.org/10.7554/eLife.40429.004

The following figure supplement is available for figure 2:

**Figure supplement 1.** Reduced number of POMC neurons in the arcuate nucleus of *Pomc*-Cre; *Dicer*$^{loxP/loxP}$ mice.

DOI: https://doi.org/10.7554/eLife.40429.005

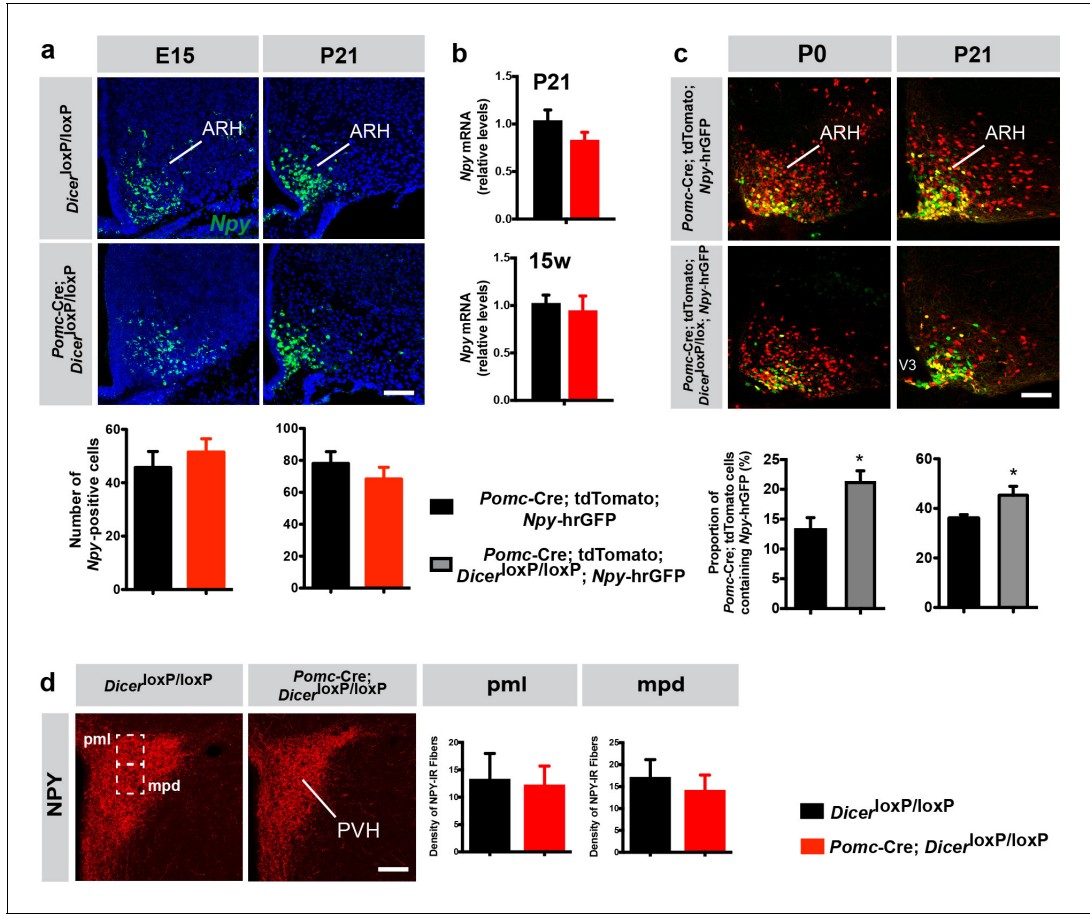

**Figure 3.** Loss of Dicer in POMC neurons favors the differentiation of *Pomc*-expressing progenitors into NPY neurons. (a) Representative images and quantification of *Npy* mRNA-expressing cells in the arcuate nucleus (ARH) of E15 (n = 5–6 per group) and P21 (n = 4 per group) *Dicer*$^{loxP/loxP}$ and *Pomc*-Cre; *Dicer*$^{loxP/loxP}$ male mice. (b) Relative expression of *Npy* mRNA in the mediobasal hypothalamus of P21 and 15-week-old *Dicer*$^{loxP/loxP}$ and *Pomc*-Cre; *Dicer*$^{loxP/loxP}$ male mice (n = 3–4 per group). (c) Representative images and quantification of *Pomc-Cre* tdTomato + cells that co-express *Npy*-hrGFP in the ARH of P0 (n = 5–6 per group) and P21 (n = 3 per group) *Pomc*-Cre and *Pomc*-Cre; *Dicer*$^{loxP/loxP}$ male mice. (d) Representative images and quantification of the density of NPY-immunopositive fibers (n = 3–4 per group) in the posterior magnocellular (pml) and medial parvicellular (mpd) parts of the paraventricular nucleus (PVH) of P21 *Dicer*$^{loxP/loxP}$ and *Pomc*-Cre; *Dicer*$^{loxP/loxP}$ male mice. Scale bars, 100 μm (a, c, d). Values are shown as mean ± SEM. *p≤0.05 *versus Pomc*-Cre; tdTomato; *Npy*-hrGFP (c). Statistical significance was determined using 2-tailed Student's *t* test. V3, third ventricle.

DOI: https://doi.org/10.7554/eLife.40429.006

levels in the hypothalamus were comparable between *Pomc*-Cre; *Dicer*^loxP/loxP^ and *Dicer*^loxP/loxP^ mice (*Figure 3b*). We also performed immunohistochemical analyses and found no difference in the density of NPY-immunoreactive fibers between control and mutant mice at P21 (*Figure 3d*). Together, these results support the hypothesis that lack of *Dicer* does not cause death of *Pomc* progenitors that subsequently adopt a NPY phenotype.

To determine if miRNAs influence the differentiation of *Pomc* progenitor cells into a NPY phenotype, we counted the number of *Pomc*-expressing progenitor cells that acquire a NPY phenotype by crossing *Pomc*-Cre; tdTomato; *Npy*-hrGFP mice with *Dicer*^loxP/loxP^ mice. At birth (P0) and P21, the proportion of *Pomc*-expressing progenitor cells that became NPY neurons was 1.6- and 1.3-fold increased, respectively, in mutant compared to control mice (*Figure 3c*). These data indicate that loss of miRNA maturation in POMC neurons favors the differentiation of *Pomc* progenitor cells into mature NPY neurons.

## Identification of miRNAs uniquely expressed in *Pomc* progenitors that become NPY neurons

To identify miRNAs that might be required for the differentiation of *Pomc* progenitors into POMC or NPY neurons, we performed 'miRNome' expression profiling using miRNA microarrays on RNA from E15 *Pomc*-eGFP cells (*i.e.*, *Pomc* progenitors that kept a POMC phenotype) (*Figure 4a*) and from E15 *Pomc*-Cre; tdTomato; *Npy*-hrGFP cells (*i.e.*, *Pomc* progenitors that acquired a NPY phenotype) (*Figure 4b*) sorted by flow cytometry. We specifically studied embryos at E15 because this age represents an important developmental period for the differentiation of *Pomc* progenitor cells into either a POMC or a NPY phenotype (*Padilla et al., 2010*). As expected, GFP-positive cells sorted from *Pomc*-eGFP hypothalami contained high levels of *Pomc* mRNA and low levels of *Npy* mRNA (*Figure 4c and d*). In contrast, GFP-positive cells sorted from *Npy*-hrGFP hypothalami and GFP +tdTomato cells sorted from *Pomc*-Cre; tdTomato; *Npy*-hrGFP embryos contained high levels of *Npy* mRNA and very low levels of *Pomc* mRNA (*Figure 4c and d*; *Figure 4—figure supplement 1a*). Out of 722 miRNA-specific probe sets, expression of 46 miRNAs was significantly changed in E15 *Pomc*→NPY cells compared with E15 *Pomc*→POMC cells, of which 22% were increased and 78% were decreased (*Figure 4e and f*; *Figure 4—figure supplement 1b*; *Supplementary file 1*). Particularly, this screen revealed a marked reduction in the expression of miR-103 and miR-107, two miRNAs that have been previously associated with obesity-induced insulin resistance in the liver and adipose tissue (*Trajkovski et al., 2011*). Using RT-qPCR, we confirmed that miR-103 and miR-107 levels were 4.0- to 4.5-fold down-regulated in E15 *Pomc*→NPY cells compared with E15 *Pomc*→POMC cells (*Figure 4g*). Moreover, a 3.2- and 1.4-fold reduction in miR-103 and miR-107 expression, respectively, was observed in sorted *Pomc* cells of *Pomc*-Cre; *Dicer*^loxP/loxP^ mice (*Figure 4—figure supplement 1c*). These data indicate that miR-103/107 might play a key role in determining whether a *Pomc* progenitor acquires a POMC or a NPY phenotype.

## In vitro silencing of miR-107 influences *pomc* progenitor specification into NPY neurons

To examine the impact of miR-103 and miR-107 in *Pomc* and *Npy* gene expression, we used antagomirs against miR-107 (Ant-107), a class of oligonucleotides that silence endogenous miRNAs (*Krützfeldt et al., 2005*). We injected these antagomirs directly into the ventricular cavity of wild-type mouse embryos in utero at E12. RT-qPCR analysis 72 hr post-injection revealed that Ant-107 caused a 2.2-fold decrease in miR-107 expression (*Figure 4h*). As previously reported (*Trajkovski et al., 2011*), Ant-107 also down-regulated miR-103 expression (data not shown). No correlation was observed between *Pomc* mRNA and miR-107 expression (*Figure 4i*). However, in embryos that displayed a knockdown efficiency greater than 60%, we found that blocking miR-107 caused a 2.4-fold decrease in *Pomc* mRNA expression (*Figure 4j*). In addition, there was a negative correlation between *Npy* mRNA expression and miR-107 expression (*Figure 4k*). When embryos displayed a knockdown efficiency greater than 60%, there was a 3.9-fold increase in *Npy* mRNA expression compared to control (*Figure 4l*).

We then established an in vitro model to study the role of miR-107 in *Pomc* progenitor differentiation. The hypothalami of *Pomc*-Cre; tdTomato; *Npy*-hrGFP embryos were dissected at E12 and prepared for incubation in a hanging drop setup (*Kredo-Russo and Hornstein, 2011*). These

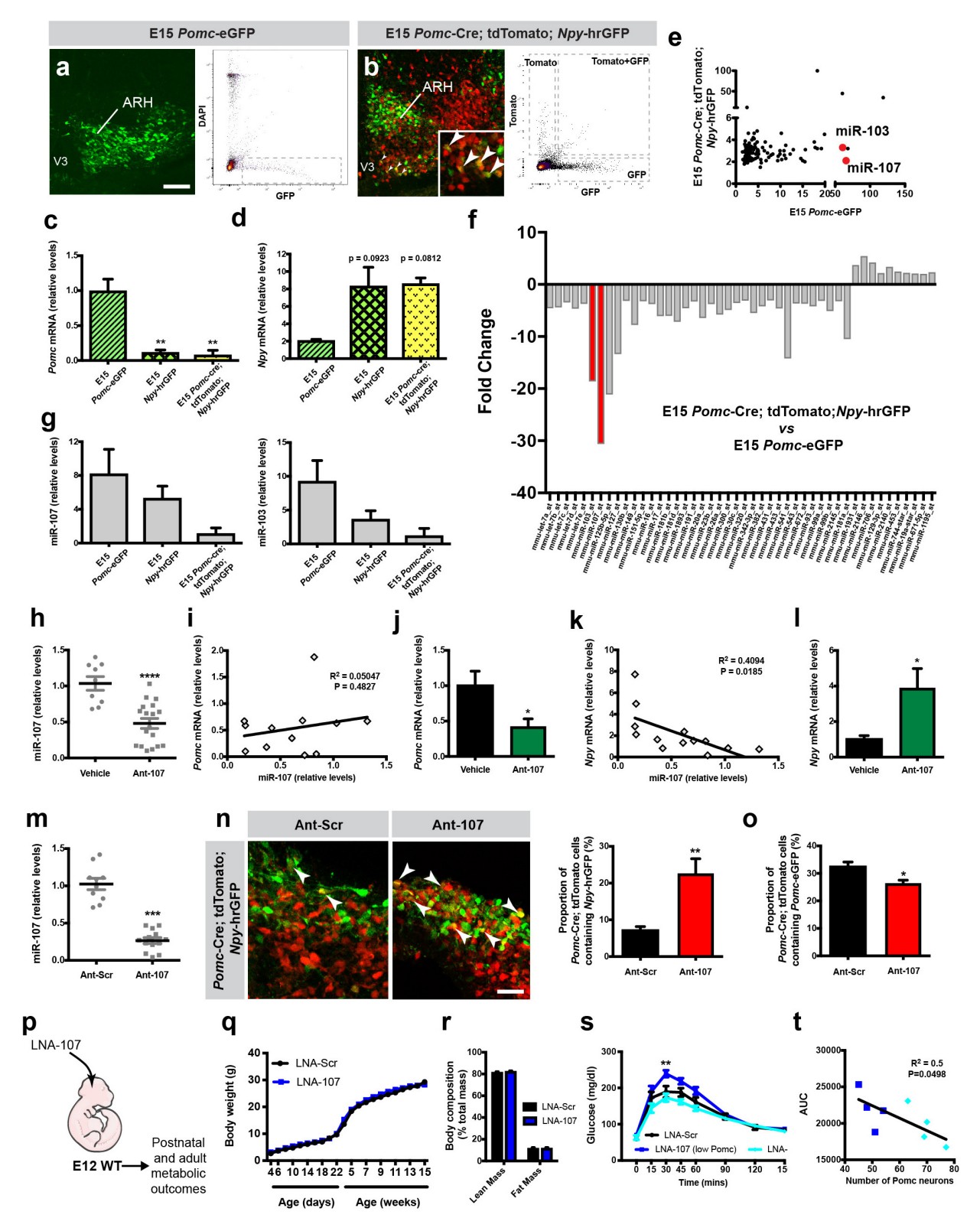

**Figure 4.** miR-107 silencing modulates *Pomc* and *Npy* expression and impairs long-term glucose homeostasis. Images and FACS isolation of (**a**) *Pomc*-eGFP[+] and (**b**) *Pomc*-tdTomato[+]/*Npy*-hrGFP[+] cells from E15 hypothalami (n = 43 – 53 per group). Arrow heads point to double-labeled cells. Relative expression of (**c**) *Pomc,* (**d**) *Npy,* and (**g**) *miR-103 and miR-107* mRNA in sorted *Pomc*-eGFP, *Npy*-hrGFP, and *Pomc*-Cre; tdTomato; *Npy*-hrGFP cells at E15 (n = 2 – 4 per group). (**e**) Scatterplots and (**f**) histograms showing the miRNA fold change in E15 *Pomc*-tdTomato/*Npy*-GFP[+] cells compared with
*Figure 4 continued on next page*

*Figure 4 continued*

E15 *Pomc*-eGFP$^+$ cells (n = 1 per group, pool of 4 – 7 samples). Relative expression of hypothalamic (h) *miR-107*, (j) *Pomc*, and (l) *Npy* mRNA in E15 embryos injected with antagomirs against miR-107 (Ant-107) or vehicle at E12 (n = 5 – 18 per group). Correlation between hypothalamic (i) *Pomc* and (k) *Npy* mRNA expression and *miR-107* expression in E15 embryos injected with Ant-107 at E12 (n = 10 – 12 per group). (m) Relative expression of miR-107 mRNA in hypothalamic embryonic explants incubated with Ant-107 or control Ant-Scr (n = 10 – 13 per group). (n) Images and quantification of *Pomc-Cre* tdTomato + cells that express *Npy*-hrGFP in hypothalamic embryonic explants incubated with Ant-107 or control Ant-Scr (n = 5 – 6 per group). (o) Quantification of *Pomc-Cre* tdTomato + cells that express *Pomc*-eGFP in hypothalamic embryonic explants incubated with Ant-107 or control Ant-Scr (n = 3 – 5 per group). (p) Experimental overview. (q) Pre- and post-weaning growth curves (body weights) of male mice injected with locked nucleic acids against miR-107 (LNA-107) or scrambled control LNA (LNA-Scr) at E12 (n = 4 – 10 per group). (r) Body composition of 15-week-old male mice injected with LNA-107 or control LNA-Scr at E12 (n = 7 – 10 per group). (s) Glucose tolerance test of 10-week-old male mice injected with LNA-107 or control LNA-Scr at E12 (n = 4 – 6 per group). Dark blue line represents animals with more than 28% reduction in the number of POMC neurons. (t) Correlation between the glucose area under the curve during GTT and number of POMC neurons in 10-week-old male mice injected with LNA-107 at E12 (n = 8 animals). Dark blue squares represent animals with more than 28% reduction in the number of POMC neurons. Scale bars, 100 µm (a, b) and 40 µm (n). Data are presented as mean ±SEM. *p≤0.05 *versus* vehicle (j, l), *versus* Ant-Scr (o); **p≤0.01 *versus* E15 *Pomc*-eGFP (c), *versus* Ant-Scr (n), *versus* LNA-Scr (s); ***p≤0.001 *versus* Ant-Scr (m); ****p≤0.0001 *versus* vehicle (h). Statistical significance was determined using 2-tailed Student's *t* test (h, j, l–o), linear regression (i, k, t), 1-way ANOVA followed by Tukey's *post hoc* test (c, d, g), and 2-way ANOVA followed by Bonferroni's *post hoc* test (q, r, s). ARH, arcuate nucleus of the hypothalamus; V3, third ventricle.

DOI: https://doi.org/10.7554/eLife.40429.007

The following figure supplement is available for figure 4:

**Figure supplement 1.** Identification and manipulation of miR-107 expression in POMC neurons.

DOI: https://doi.org/10.7554/eLife.40429.008

organotypic explants were then incubated with Ant-107 or scrambled control antagomirs against miR-690 (Ant-Scr). After 72 hr, the addition of the Ant-107 to the culture medium produced a 4.9-fold reduction in miR-107 expression (*Figure 4m*). This reduction in miR-107 expression was accompanied with a 3.2-fold increase in the proportion of *Pomc* progenitors that differentiate into NPY neurons (*Figure 4n*). Interestingly, the silencing of miR-107 is also accompanied with a 1.2-fold reduction in the number of *Pomc*-expressing progenitors that express *Pomc*-eGFP (*Figure 4o*). These observations support the hypothesis that a reduction in miR-107 expression favors the differentiation of *Pomc* progenitors into a NPY phenotype *versus* a POMC phenotype.

## In utero silencing of miR-103/107 has long-term effects on glucose homeostasis

The data described above support a role for miR-103/107 in the peptidergic differentiation of *Pomc* progenitor cells during embryonic life. To further explore the importance of prenatal miR-107 in life-long metabolic regulation, we synthesized locked nucleic acids that specifically target the seed sequence of miR-107 and inhibit its expression (LNA-107), injected them into the ventricular cavity of wild-type mouse embryos at E12. We observed a 2-fold reduction in miR-107 expression in the hypothalami of E14 embryos that received LNA-107 at E12 compared to control mice (*Figure 4p*; *Figure 4—figure supplement 1d*). Physiologically, the pre- and post-weaning growth curves (body weights) of injected mice injected with LNA-107 prenatally were undistinguishable from those of control mice (*Figure 4q*). Similarly, body composition was similar between LNA-107-injected and control mice (*Figure 4r*). However, when exposed to a glucose challenge, adult mice that received LNA-107 injections prenatally and had 34% loss in POMC neurons displayed impaired glucose tolerance compared with control mice (*Figure 4s*). In addition, there was a negative correlation between the glucose area under the curve during GTT and the number of POMC neurons (*Figure 4t*).

## Discussion

Several lines of evidence have indicated that small non-coding RNAs or miRNAs are involved in the regulation of metabolism and neural development (*Coolen and Bally-Cuif, 2009*; *Oliverio et al., 2016*; *Fineberg et al., 2009*; *Vo et al., 2010*; *Deiuliis, 2016*; *McGregor and Choi, 2011*; *Schneeberger et al., 2012*; *Jordan et al., 2011*; *Kornfeld et al., 2013*). However, whether miRNAs play a role in the development of CNS pathways that control energy balance remains elusive. In the present study, we report that loss of *Dicer*, an enzyme critical to process miRNA precursors (pre-miRNAs) into mature miRNAs (miRNAs), in POMC neurons causes metabolic disturbances in mice

associated with an age-related decline in the number of *Pomc* mRNA-expressing cells. We also show that *Dicer* influences the peptidergic differentiation of *Pomc* progenitor cells during embryonic life, a developmental process that appears to be specifically mediated through miR-103/107.

A marked reduction in the number of *Pomc* mRNA-expressing cells was observed in the absence of *Dicer*. The number of neurons is a function of several developmental processes including cell proliferation, migration, death but also cell differentiation. The majority of neurons located in the mouse hypothalamus, including POMC neurons, are born between E11 and E12 (*Ishii and Bouret, 2012*; *Padilla et al., 2010*). The observation that number of *Pomc* mRNA-expressing cells is reduced as early as at E13 suggests that neurogenesis might be impaired in *Pomc*-Cre; *Dicer*^loxP/loxP mice, as previously reported in other brain regions such as the cortex and spinal cord (*Kawase-Koga et al., 2009*). The distribution pattern of POMC neurons in mutant mice resembles that of control mice, supporting the idea that Dicer does not influence POMC cell migration. Our neuroanatomical analysis also revealed that Dicer deficiency causes a marked reduction in the number of *Pomc* mRNA-expressing cells particularly after birth and that 90% of POMC⁺ cells are loss in the arcuate nucleus of adult *Pomc*-Cre; *Dicer*^loxP/loxP mice. This dramatic reduction in the number of *Pomc* mRNA-expressing neurons might be the result of increased cell death consistent with previous papers implicating *Dicer* and miRNAs in cell survival of various neuronal systems including the retina, the neural crest, and the hypothalamus (*Sundermeier et al., 2014*; *Zehir et al., 2010*; *Damiani et al., 2008*; *Schneeberger et al., 2012*). However, our cell lineage tracing experiments that allowed us to genetically label *Pomc*-expressing cells, independently of *Pomc* mRNA content, further revealed that a significant number of neurons survive in mutant mice and that *Pomc* mRNA expression is blunted in these neurons. Because in our mouse model Cre activity is turned on in *Pomc*-expressing cells from embryonic life to adulthood, future studies using temporal deletion of Dicer are required to investigate its relative contribution on developmental processes *versus* adult regulations.

A unique developmental property of POMC neurons during embryonic life is that they can adopt an orexigenic NPY phenotype (*Padilla et al., 2010*). Here we show that *Pomc*-Cre; *Dicer*^loxP/loxP mice display a higher number of *Pomc* progenitors that differentiate into a NPY phenotype. We also report that this developmental process involves miR-103/107. These data are in agreement with previous findings showing that miRNAs can be involved in cell fate determination during brain development (*Fazi and Nervi, 2008*). For example, the transcriptional repressor RE1 silencing transcription factor interacts with miR-124a to maintain neuronal progenitors in a non-neuronal cells lineage. When progenitors differentiate in mature neurons, REST leaves the miR-124a loci enabling miR-124a to repress non-neuronal transcripts (*Conaco et al., 2006*). Our in vivo experiments indicate that miR-103/107 regulate *Pomc* and *Npy* gene expression. The exact molecular mechanisms underlying the effect of miR-103/107 on *Pomc* and *Npy* mRNA levels remain to be studied as these miRNAs do not directly target POMC or NPY 3'-UTRs. Transcription factors such as neurogenin three and Mash1 have been shown to be involved in the neurogenesis and the specification of POMC and NPY neurons (*Pelling et al., 2011*; *McNay et al., 2006*). However, predicting target gene analysis revealed that these transcription factors are not direct targets for miR-103/107. Nevertheless, the present study represents the first demonstration that miR-103/107 might influence hypothalamic development. It is also the first study that analyzes the molecular mechanisms that underlie the timely differentiation of *Pomc* progenitors into a POMC *versus* NPY phenotype. It remains possible that the loss of *Dicer* and miR-103/107 might also result in the differentiation of *Pomc* progenitors into other phenotypes. For example, Sanz *et al.* have shown that a sub-population of *Pomc*-expressing progenitors can also differentiate into kisspeptin and tachykinin neurons (*Sanz et al., 2015*).

The role of miR-103/107 in metabolic regulation has mainly been studied in the periphery. miR-103/107 control glucose homeostasis and are upregulated in the liver of obese mice (*Trajkovski et al., 2011*). Silencing miR103/107 in the liver and fat improves glucose homeostasis and insulin sensitivity (*Trajkovski et al., 2011*). In contrast, gain of miR-103/107 function in either the liver or adipose tissue induces impaired glucose homeostasis (*Trajkovski et al., 2011*). Moreover, overexpression of miR-107 induces endoplasmic reticulum stress in the liver and causes lipid accumulation in adipocytes (*Bhatia et al., 2014*). In addition, Vinnikov and colleagues reported that intra-ARH delivery of miR-103 mimics ameliorate hyperphagia and obesity in mice lacking Dicer in forebrain neurons (*Vinnikov et al., 2014*). However, this study did not examine glucose homeostasis and mimic injections were performed in adult mice. Our data indicate that blockade of miR-107 in the prenatal brain affects glucose homeostasis with no significant effect on body weight or body

composition. These findings contrast with the phenotype of *Pomc*-Cre; *Dicer*$^{loxP/loxP}$ mice that display obesity in addition to perturbations in glucose homeostasis. These observations indicate that miRNAs other than miR-103/107 might be involved in the central regulation of body weight and adiposity. For example, miR-155 has recently been suggested to play a role in the central regulation of body weight and food intake (*Maldonado-Avilés et al., 2018*). In any events, by combining the use of miRNA manipulation with cell lineage, and physiological experiments, the present study reveals a novel role for miR-103/107 in determining the ultimate peptidergic phenotype of hypothalamic neurons involved in glucose regulation.

## Materials and methods

### Animals

All experimental procedures were approved by the Institutional Animal Care and Use Committee of Children's Hospital of Los Angeles (protocol #303 – 16). Mice were grouped-housed in individual cages under specific pathogen-free conditions, maintained in a temperature-controlled room with a 12 hr light/dark cycle, and provided *ad libitum* access to water and standard laboratory chow (Special Diet Services). *Pomc*-Cre mice (Jax mice stock# 5965) (*Balthasar et al., 2004*) were mated to *Dicer*$^{loxP/loxP}$ mice (Jax mice stock# 6001) (*Harfe et al., 2005*) to generate *Pomc*-specific DICER knockout (*Pomc*-Cre; *Dicer*$^{loxP/loxP}$). These mice were then crossed with a ROSA-tdTomato reporter line (Jax mice stock# 7914) (*Madisen et al., 2010*) so that neurons that express *Pomc* from gestation are permanently marked. This triple transgenic mouse model was also crossed with mice that selectively express humanized renilla GFP (hrGFP) in *Npy*-containing neurons (generously provided by Dr. Bradford Lowell) to facilitate the visualization of *Pomc*→NPY neurons. Homozygous transgenic mice that selectively express an enhanced GFP (eGFP) in *Pomc*-containing neurons were kindly provided by Dr. Malcolm J. Low. CD-1 and C57Bl/6 mice (Jax mice) were also used for intra-embryonic experiments. For the studies involving embryos, the breeders were mated around 6:00 pm and checked for a vaginal plug the next day. The day of conception (sperm positive vaginal smear) was designated as embryonic day (E) zero (E0). The day of birth was considered postnatal day 0 (P0).

### MicroRNA inhibitors

The term antagomir refers to cholesterol-conjugated modified RNA (*Krützfeldt et al., 2005*). Oligonucleotides used in this study consisted of 22 or 23 nucleotides with followed modifications. mmu-miR-690 (scrambled antagomir):

5'-mU*mU*mUmGmGmUmUmGmUmGmAmGmCmCmUmAmGmCmC*mU*mU*mU*−3'  chol  ; mmu-miR-107 :

5'-mU*mG*mAmUmAmGmCmCmCmUmGmUmAmCmAmAmUmGmCmU*mG*mC*mU*−3'chol. (m = 2'-O-methyl, *=phosphorothioate).

The term LNA refers to custom-made miRCURY locked nucleic acids (LNA) for in vivo application. They were designed and synthetized as unconjugated or FAM-conjugated and contain phosphorothioate oligonucleotides (Exiqon). The sequence of the LNA targeting miR-107 (LNA-107) was complementary to the mature mmu-miR-107 – 3 p : 5'-CCTGTACAATGCTGC-3'. The conjugated scrambled LNA (LNA-Scr) sequence is the following : 5'/56-FAM/ACGTCTATACGCCCA-3'.

### Intra-embryonic injections

Timed-pregnant WT mice carrying E12 embryos were anesthetized, and the uterine horns were gently placed outside the abdominal cavity. Using a Nanofil syringe with a 35G needle attachment (World Precision Instruments), miR-107 antagomirs (Ant-107, 0.5 nmol, Dharmacon; for gene expression studies) or LNA-107 (1 nmol, Exiqon; for physiological studies) was then delivered into each embryo intracerebroventricularly. KPBS, scrambled antagomirs (Ant-Scr) or LNA (LNA-Scr) were used as control. After injections, the uterus was carefully placed back into the pregnant mother and the incision was closed with sutures. Each experiment included offspring from at least three litters.

### Hanging drop tissue culture

The hypothalami of *Pomc*-Cre; tdTomato; *Pomc*-eGFP (n = 8 embryos from two independent experiments) and *Pomc*-Cre; tdTomato; *Npy*-hrGFP E12 embryos (n = 13 embryos from four independent

experiments) were microdissected under a stereomicroscope and prepared for incubation in a hanging drop setup as previously described (*Kredo-Russo and Hornstein, 2011*). Explants were incubated in fresh modified Basal Medium Eagle (Invitrogen) containing antagomirs against miR-107 or miR-690 (scrambled control) (0.2 nmol, Dharmacon). After 3 days in vitro, the explants were fixed in 4% paraformaldehyde, frozen and sectioned.

## Tissue collection

Hypothalami of E12, E14, E16 and mediobasal hypothalami of P10, P21 and 15 week-old *Dicer*$^{loxP/loxP}$ and *Pomc*-Cre; *Dicer*$^{loxP/loxP}$ mice (n = 3 – 5 per group) were microdissected under a stereomicroscope.

## Cell sorting

Hypothalami of E13 *Pomc*-eGPP (41 embryos from six litters), E15 *Pomc*-eGPP (53 embryos from seven litters), E15 *Npy*-hrGFP (46 embryos from six litters), E15 *Pomc*-Cre; tdTomato; *Npy* -hrGFP (43 embryos from five litters) and P21 *Pomc*-Cre; tdTomato; *Dicer*$^{loxP/loxP}$ mice and *Pomc*-Cre; tdTomato mice (n = 4 per group) were microdissected under a stereomicroscope and enzymatically dissociated using the Papain Dissociation System (Worthington) following the manufacturer's instructions. Fluorescence-Activated Cell Sorting (FACS) was performed using a BD FACS Aria II Cell Sorter to sort *Pomc*-eGFP +cells, *Npy*-hrGFP +cells and cells containing both *Pomc*-Cre; tdTomato and *Npy*-hrGFP. Non-fluorescent cells obtained from wild-type mice were used set the threshold for fluorescence.

## RT-qPCR analyses

RNA containing the miRNA fraction was then extracted using the MiRVana miRNA isolation kit (Ambion) according to the manufacturer's protocol.

For miRNA expression analyses, cDNA was prepared from 10 ng total RNA using TaqMan MicroRNA Reverse Transcription Kit (Life Technologies). qRT-PCR was performed using TaqMan Universal PCR Mastermix No AmpErase UNG and the commercially available Taqman MicroRNA primers for *hsa-miR-103* (000439), *hsa-miR-107* (000443) and *hsa-miR-690* (001677). *Sno135* (001230) or *U6snRNA* (001973) were used for endogenous control.

For gene expression analyses, cDNA was generated with the high-capacity cDNA Reverse Transcription kit (Life Technologies). qRT-PCR was performed using TaqMan Fast Universal PCR Mastermix and the commercially available Taqman gene expression primers for *Pomc* (Mm00435874_m1), *Npy* (Mm03048253_m1), and *Dicer* (Mm00521722_m1). *Gapdh* (Mm99999915_g1) was used for endogenous control.

All assays were performed using an Applied Biosystems 7900 HT real-time PCR system. Calculations were performed by a comparative method ($2-\Delta\Delta CT$).

## Microarray analysis

six samples were pooled for E15 *Npy*-hrGFP +cells, seven samples were pooled for E15 *Pomc*-eGFP +cells and four samples for Pomc-Cre; tdTomato; Npy-hrGFP +cells. 125 ng of total RNA for each sample obtained from sorted cells was labeled using the FlashTag Biotin RNA Labeling kit (Genisphere) and hybridized to a GeneChip miRNA 2.0 array (Affymetrix) according to the manufacturer's instructions. Affymetrix miRNA array chips identify miRNAs and pre-miRNAs in 131 organisms. Content is derived from Sanger miRbase miRNA database V15. Raw data files were imported into the Partek Genomic Suite 6.6 beta software (USC) and signal intensities for probe sets were quantile- normalize by robust multichip averaging (RMA) using miRNA-2_0.annotations.20101222.csv.

## Tissue preparation

P0, P10, P14, P21 and 15-week-old were anesthetized and perfused transcardially with 4% paraformaldehyde (n = 3 – 6 per group). For embryonic tissue, dams were anesthetized and E13 and E15 embryos (n = 5 – 8 per group) were dissected in cold PBS and fixed 4% paraformaldehyde. The brains were then frozen and 20- (for embryos and P0, P10 and P14 mice) or 30-um-thick coronal sections (for P21 and adults) were performed and collected on slides.

## Immunohistochemistry

Sections were processed for immunofluorescence using standard procedures (*Bouret et al., 2004b*; *Steculorum et al., 2015*; *Croizier et al., 2016*). The primary antibodies used for IHC were as follows: sheep anti-α-MSH (1:40,000, Millipore; *Steculorum et al., 2015*), sheep anti-NPY (1:3,000, Chemicon; *Lee et al., 2013*), rabbit anti-β-endorphin (1:5,000, Millipore; *Grayson et al., 2010*), rabbit anti-POMC (1:2,000, Phoenix Pharmaceuticals; *MacKay et al., 2017*). The primary antibodies were visualized with Alexa Fluor 568 goat anti-rabbit IgGs or Alexa Fluor 568 donkey anti-sheep IgGs (1:200, Invitrogen). Sections were counterstained using bisbenzamide (1:10,000, Invitrogen), to visualize cell nuclei, and coverslipped with buffered glycerol (pH 8.5).

## Fluorescent in situ hybridization

Sense and antisense digoxigenin-labeled riboprobes were generated from plasmids containing PCR fragments of *Npy* and *Pomc*. Sections were post-fixed in 4% paraformaldehyde and incubated with Proteinase K (Promega). They were incubated in triethanolamine (TEA) and then in TEA containing glacial acid acetic. Sections were pre-hybridized in hybridization buffer and then hybridized with denatured probes (300 ng for *Npy* and 400 ng for *Pomc*) overnight at 62 C. After washes in stringency solutions, sections were blocked in TNB solution (Roche) and incubated in a horseradish peroxidase-conjugated sheep anti-DIG antibody (1:400, Roche Applied Sciences; *Padilla et al., 2010*). DIG was visualized using a TSA PLUS Biotin Kit (Perkin Elmer). Sections were first incubated in the Biotin Amplification Reagent (1:50), and then in cyanin 2-conjugated streptavidin (1:200, Jackson Immunoresearch). Sections were counterstained using bisbenzamide (1:10,000, Invitrogen) to visualize cell nuclei.

## Image analysis

Images were acquired using a Zeiss LSM 710 confocal system equipped with a 20X objective through the ARH and NTS (for cell number and cell fate specification) and through the pml and mpd parts of the PVH (for axonal projections). The average number of cells and density of fibers were analyzed in 2 – 4 hemisections.

For the quantitative analysis of cell number, four categories of labeled cells were manually counted using Image J software: (1) numbers of POMC$^+$ cells; (2) numbers of *Pomc*-tdTomato$^+$ cells; (3) numbers of NPY$^+$ cells; and (4) numbers of *Pomc*-tdTomato + *Npy* hrGFP double-labeled cells. Only cells with corresponding bis-benzamide-stained nucleus were included in our counts.

Quantitative analysis of fiber density was performed using Image J analysis software (NIH) as described (*Bouret et al., 2008*; *Coupé et al., 2012*). Each image plane was binarized to isolate labeled fibers from the background and to compensate for differences in fluorescence intensity. The integrated intensity was then calculated for each image as previously described. This procedure was conducted for each image plane in the stack, and the values for all of the image planes in a stack were summed. Three categories of labeled fibers were quantified: (1) relative density of β-endorphin$^+$ fibers; (2) relative density of α-MSH$^+$ fibers; (3) relative density of NPY$^+$ fibers.

## Physiological measures

Male and female mice (n = 7 – 9 per group) were weighed every 2 days from P4 to P21 (weaning) and weekly from 4 to 15 weeks using an analytical balance. To measure food consumption, 13-week-old mice (n = 5 – 7 per group) were housed individually in cages, and, after 1 day of acclimation, food intake was measured every 12 hr for 3 days from pre-weighed portions of food dispensed from the wire cage tops. The cumulative food intake of each mouse (9- and 16-week-old mice; n = 5 – 6 per group) was used for statistical comparisons. Body composition analysis (fat/lean mass) was performed in 16-week-old mice (n = 4 per group) using NMR (Echo MRI). Perigonadal fat depots were collected in 15-week-old mice (n = 4 – 5 per group) and weighed. Respiratory exchange ratio (RER), energy expenditure and locomotor activity was monitored at 15 weeks of age using a combined indirect calorimetry system (TSE systems). Briefly, after adaptation for 1 week, $O_2$ and $CO_2$ production was measured for 4 days to determine the RER and energy expenditure (n = 4 – 6 per group) on composition-matched mice. Glucose tolerance test (GTT) was conducted in 10- to 11-week-old mice (n = 7 – 9 per group) through i.p. injection of glucose (1.5 mg/g body weight) after overnight fasting. Blood glucose levels were measured at 0, 15, 30, 45, 60, 90, 120, and 150 min post-injection.

Leptin sensitivity tests were performed in 12- to −13 week-old male mice (n = 5 per group). Briefly, mice were injected i.p. with vehicle (5 mM sodium citrate buffer) or leptin (3 mg/kg body weight, Peprotech) according to the following scheme: vehicle injections for 3 days, followed by leptin injections for 3 days. Body weight was measured during the injection period. Serum leptin and insulin levels were assayed at 15 and 20 – 23 weeks of age (n = 5 – 6 per group) using leptin and insulin ELISA kits, respectively (Millipore).

## Statistical analysis

All values were represented as the mean ±SEM. Statistical analyses were conducted using GraphPad Prism (version 5.0a). For each experiment, slides were numerically coded to obscure the treatment group. Statistical significance was determined using unpaired 2-tailed Student's t test, 1-way ANOVA followed by Tukey *post hoc* test, 2-way ANOVA followed by the Bonferroni *post hoc* test, and linear regression when appropriate. $p \leq 0.05$ was considered statistically significant.

## Acknowledgments

We thank the CHLA Rodent Metabolic Core for metabolic studies and the CHLA Genomic Core for cell sorting and microarray studies. We are also grateful to Gricelda Vasquez with her assistance with animal husbandry. This work was supported by the National Institutes of Health (Grants DK84142, DK102780, and DK118401 to SGB).

## Additional information

### Funding

| Funder | Grant reference number | Author |
| --- | --- | --- |
| National Institute of Diabetes and Digestive and Kidney Diseases | DK84142 | Sebastien G Bouret |
| National Institute of Diabetes and Digestive and Kidney Diseases | DK102780 | Sebastien G Bouret |
| National Institute of Diabetes and Digestive and Kidney Diseases | DK118401 | Sebastien G Bouret |

The funders had no role in study design, data collection and interpretation, or the decision to submit the work for publication.

### Author contributions

Sophie Croizier, Conceptualization, Data curation, Formal analysis, Validation, Methodology, Writing—original draft, Conceived, designed and performed most of the experiments, Analyzed data; Soyoung Park, Data curation, Formal analysis, Performed and analyzed POMC immunohistochemistry on E15 embryo pituitary and RT-qPCRs required for revisions; Julien Maillard, Data curation, Formal analysis, Performed and analyzed some RT-qPCR experiments; Sebastien G Bouret, Conceptualization, Formal analysis, Supervision, Funding acquisition, Validation, Investigation, Methodology, Project administration, Writing—review and editing, Conceived, designed and supervised the project

### Author ORCIDs

Sophie Croizier (ID) https://orcid.org/0000-0002-0076-1008
Sebastien G Bouret (ID) https://orcid.org/0000-0002-4174-9769

### Ethics

Animal experimentation: CHLA IACUC protocol #303-16

Decision letter and Author response
Decision letter https://doi.org/10.7554/eLife.40429.012
Author response https://doi.org/10.7554/eLife.40429.013

## Additional files

### Supplementary files

• Supplementary file 1. List of miRNAs up- and down-regulated in sorted *Pomc*-Cre; tdTomato; *Npy*-hrGFP cells *versus* E15 *Pomc*-eGFP cells derived from E15 embryos
DOI: https://doi.org/10.7554/eLife.40429.009

• Transparent reporting form
DOI: https://doi.org/10.7554/eLife.40429.010

### Data availability

All data generated or analysed during this study are included in the manuscript and supporting files.

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
