## [Decision Letter]

Thank you for submitting your article "*Dicer*-miR-103/107 controls developmental switch of POMC progenitors into NPY neurons and impacts glucose homeostasis" for consideration by *eLife*. Your article has been reviewed by three peer reviewers and the evaluation has been overseen by a Reviewing Editor and a Senior Editor. The following individual involved in review of your submission has agreed to reveal his identity: Jens Claus Brüning (Reviewer #2).

The reviewers have discussed the reviews with one another and the Reviewing Editor has drafted this decision to help you prepare a revised submission.

Summary:

Croizier and colleagues describe results from studies investigating the role of *Dicer* and miR103/107 in POMC neurons to regulate energy balance, glucose homeostasis, and POMC expression. They used a number of approaches to provide evidence that miR 103/107 play a role in maintaining POMC expression, ultimately affecting glucose homeostasis and energy balance. The have also identified a role for miRNA-dependent regulation during the process specifying NPY neurons, which develop from POMC-positive precursors. Overall, the studies are well described and provide novel observations that will be of wide interest. A few issues need to be addressed.

Essential revisions:

The authors provide extensive evidence that *Dicer* (and likely miR103/107) are affecting POMC expression in the hypothalamus. However, unless I missed it, the authors did not assess if POMC expressing cells in the anterior pituitary were also affected. Clearly, altered activity of the HPA axis could help contribute to the complex phenotypes observed. Similarly, the reason for no effects on POMC neuron number in hindbrain is not clear.

A comparative miRNA expression profile between POMC/POMC and POMC/NPY neurons was performed. The authors found differential regulation of miR103/107 expression, miRNAs, which had previously been linked to insulin sensitivity in liver. Through a combination of in vitro and in vivo studies, they predict that these miRNAs indeed regulate the propensity of POMC neurons to develop into NPY-expressing cells and that ARH expression of these miRNAs in utero affects metabolism in the offspring.

However, no data was presented to show that deletion of *Dicer* in POMC caused specific changes in miR103/107. At a minimum, a discussion on potential contribution from changes in other miRNAs to the phenotypes of POMC *Dicer* deletion may be necessary. More helpful would be direct assessment in the *Dicer* knockouts.

The topic on POMC transition to NPY by miRNAs is interesting; however, this transition only accounts for a very minor number among those "lost" POMC neurons. A discussion on the potential fate of the majority of other, the majority of "lost" POMC neurons will reduce potential confusion to readers.

Also, it would be quite helpful for the authors to clearly state that the decrease is in POMC detectability vs. cell loss. The authors should clarify this.

Given the wide readership of *eLife*, it would be helpful if the authors provide a couple of comments about inherent limitations of doing the prenatal deletion of *Dicer*. Also, the potential role of developmental effects vs. adult role of *Dicer* in POMC cells could be commented upon.

It is unclear if all the metabolic cage studies were done in weight/composition matched mice. If so, please state clearly.

---

## [Author Response]

Essential revisions:The authors provide extensive evidence that Dicer (and likely miR103/107) are affecting POMC expression in the hypothalamus. However, unless I missed it, the authors did not assess if POMC expressing cells in the anterior pituitary were also affected. Clearly, altered activity of the HPA axis could help contribute to the complex phenotypes observed.

We thank the reviewers for this interesting suggestion. We have now added data on POMC-immunopositive cell numbers in the anterior pituitary of *Dicer*^loxP/loxP^ and *Pomc-*Cre; *Dicer*^loxP/loxP^ mice (Results secttion; Materials and methods section; new Figure 2—figure supplement 1E).

Similarly, the reason for no effects on POMC neuron number in hindbrain is not clear.

We agree with the reviewers that these observations can be surprising. However, since our quantification of the number of POMC neurons in the hindbrain are derived from genetically labeled *Pomc* neurons (using *Pomc-*Cre; tdTomato mice), it remains possible that the number of aMSH/POMC-immunopositive cells might be different between control and mutant mice.

*A comparative miRNA expression profile between POMC/POMC and POMC/NPY neurons was performed. The authors found differential regulation of miR103/107 expression, miRNAs, which had previously been linked to insulin sensitivity in liver. Through a combination of* in vitro *and* in vivo studies, they predict that these miRNAs indeed regulate the propensity of POMC neurons to develop into NPY-expressing cells and that ARH expression of these miRNAs in utero affects metabolism in the offspring.However, no data was presented to show that deletion of Dicer in POMC caused specific changes in miR103/107. At a minimum, a discussion on potential contribution from changes in other miRNAs to the phenotypes of POMC Dicer deletion may be necessary. More helpful would be direct assessment in the Dicer knockouts.

We agree that this important data needs to be added in the manuscript. Accordingly, we quantified the level of miR-103 and miR-107 expression in sorted POMC cells derived from *Pomc-*Creand *Pomc-*Cre; *Dicer*^loxP/loxP^ mice (Results section; Materials and methods section; new Figure 4—figure supplement 1C).

The topic on POMC transition to NPY by miRNAs is interesting; however, this transition only accounts for a very minor number among those "lost" POMC neurons.A discussion on the potential fate of the majority of other, the majority of "lost" POMC neurons will reduce potential confusion to readers.

This interesting point is now discussed in the revised manuscript (Discussion section).

Also, it would be quite helpful for the authors to clearly state that the decrease is in POMC detectability vs. cell loss. The authors should clarify this.

We apologize if this point was confusing. We have now clarified that the decrease in cell numbers observed in *Pomc-*Cre; *Dicer*^loxP/loxP^ mice is due to a reduction in POMC detectability *versus* increased cell loss (Results section; Discussion section).

Given the wide readership of eLife, it would be helpful if the authors provide a couple of comments about inherent limitations of doing the prenatal deletion of Dicer. Also the potential role of developmental effects vs. adult role of Dicer in POMC cells could be commented upon.

We agree with the reviewers that adding these limitations would be helpful

given the wide readership of *eLife*. This point is now discussed (Discussion section).

It is unclear if all the metabolic cage studies were done in weight/composition matched mice. If so, please state clearly.

We apologize for this omission. Metabolic cage studies were performed in composition matched mice. This point is now added in the Materials and methods section.